# Data Unlearning in Diffusion Models

**Silas Alberti**[*] **Kenan Hasanaliyev**[*] **Manav Shah** **Stefano Ermon**

Stanford University
`{alberti,kenanhas,manavs,ermon}@cs.stanford.edu`

## Abstract

Recent work has shown that diffusion models memorize and reproduce training data examples. At the same time, large copyright lawsuits and legislation such as GDPR have highlighted the need for erasing datapoints from diffusion models. However, retraining from scratch is often too expensive. This motivates the setting of data unlearning, i.e., the study of efficient techniques for unlearning specific datapoints from the training set. Existing concept unlearning techniques require an anchor prompt/class/distribution to guide unlearning, which is not available in the data unlearning setting. General-purpose machine unlearning techniques were found to be either unstable or failed to unlearn data. We therefore propose a family of new loss functions called Subtracted Importance Sampled Scores (SISS) that utilize importance sampling and are the first method to unlearn data with theoretical guarantees. SISS is constructed as a weighted combination between simpler objectives that are responsible for preserving model quality and unlearning the targeted datapoints. When evaluated on CelebA-HQ and MNIST, SISS achieved Pareto optimality along the quality and unlearning strength dimensions. On Stable Diffusion, SISS successfully mitigated memorization on nearly 90% of the prompts we tested. We release our code online.[1]

## 1 Introduction

The recent advent of diffusion models has revolutionized high-quality image generation, with large text-to-image models such as Stable Diffusion (Rombach et al., 2022) demonstrating impressive stylistic capabilities. However, these models have been shown to memorize and reproduce specific training images, raising significant concerns around data privacy, copyright legality, and the generation of inappropriate content (Carlini et al., 2023; Cilloni et al., 2023). Incidents such as the discovery of child sexual abuse material in LAION (Thiel, 2023; Schuhmann et al., 2022) as well as the need to comply with regulations like the General Data Protection Regulation and California Consumer Privacy Act that establish a "right to be forgotten" (Hong et al., 2024; Wu et al., 2024), underscore the urgency of developing effective methods to remove memorized data from diffusion models.

Retraining on a new dataset is often prohibitively expensive, and the bulk of traditional machine unlearning techniques have been built for classical supervised machine learning (Cao & Yang, 2015; Ginart et al., 2019; Izzo et al., 2021; Bourtoule et al., 2021). Recently, a new wave of research on unlearning in diffusion models has emerged, but it has focused almost exclusively on *concept unlearning* in text-conditional models (Gandikota et al., 2023; Kumari et al., 2023; Zhang et al., 2023; Gandikota et al., 2024; Schramowski et al., 2023; Heng & Soh, 2023; Fan et al., 2024). These approaches aim to remove higher-level concepts, e.g., the styles of painters or nudity, rather than specific datapoints from the training data needed to battle unwanted memorization. This paper focuses on the problem of efficient machine unlearning in diffusion models with the objective of removing specific datapoints, a problem that we refer to as *data unlearning*.

Unlike concept unlearning, the data unlearning setting has a concrete gold standard: retraining without the data to be unlearned. The goal of data unlearning is to achieve unlearning performance as close as possible to retraining while using less computational resources. To quantify unlearning performance,

---

[*]Equal contribution

[1]`https://github.com/claserken/SISS`

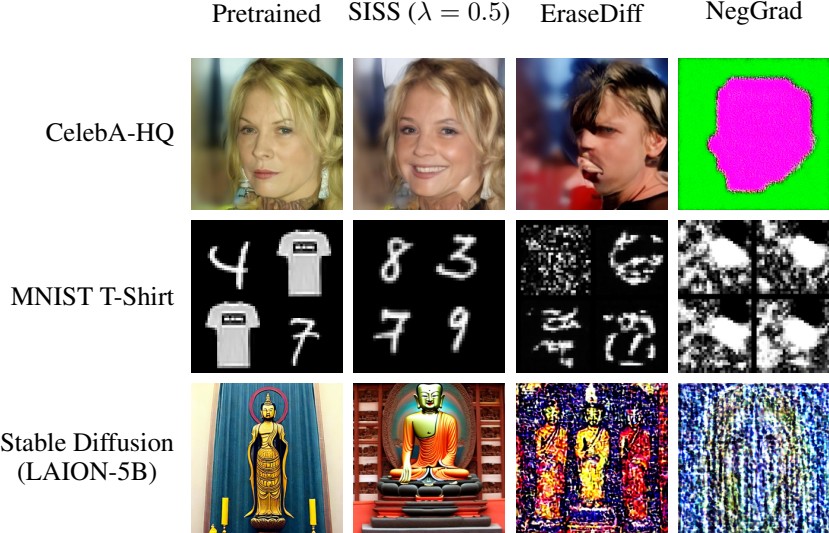

Figure 1: Examples of quality degradation across unlearning methods. On all 3 datasets, we find that our SISS method is the only method capable of unlearning specific training datapoints while maintaining the original model quality. See Tables 1, 2 and Figure 6a for complete quantitative results on quality preservation.

we focus on three separate areas: the degree of unlearning, the amount of quality degradation after unlearning, and the amount of compute needed. Examples of these areas are highlighted in Figures 1 and 2.

When applied to data unlearning, general-purpose machine unlearning techniques face certain limitations: naive deletion (fine-tuning on data to be kept) tends to be slow in unlearning, while NegGrad (gradient ascent on data to be unlearned) (Golatkar et al., 2020) forgets the data to be kept, leading to rapid quality degradation. State-of-the-art class and concept unlearning techniques do not apply to our setting because they require an anchor prompt/class/distribution to guide the unlearning towards which we do not assume access to. For example, Heng & Soh (2023) select a uniform distribution over pixel values to replace the $0$ class on MNIST; however, the desired unlearning behavior would be to instead generate digits $1 - 9$ when conditioned on $0$. Even if one selects the target distribution to be a random example/label from all other classes as Fan et al. (2024) do, it is not always clear what "all other classes" are in the text-conditional case. Furthermore, these prior experiments are targeting prompts/classes instead of datapoints and do not apply in the case of unconditional diffusion models. A notable exception is EraseDiff (Wu et al., 2024) which can unlearn data by fitting the predicted noise to random noise targets that are not associated with a prompt/class.

In this work, we derive an unlearning objective that combines the objectives of naive deletion and NegGrad. For further computational efficiency, we unify the objective through importance sampling, cutting the number of forward passes needed to compute it by half. We term our objective Subtracted Importance Sampled Scores (SISS). As seen in Figure 1, SISS allows for the computationally efficient unlearning of training data subsets while preserving model quality. It does so because the naive deletion component preserves the data to be kept, while the NegGrad component targets the data to be unlearned. The addition of importance sampling balances between the two components through a parameter $\lambda$, where $\lambda = 0$ and $\lambda = 1$ behave like naive deletion and NegGrad, respectively. We find that $\lambda = 0.5$ behaves as a mixture of the two, giving the desirable combination of quality preservation and strong unlearning.

We demonstrate the effectiveness of SISS on CelebA-HQ (Karras et al., 2018), MNIST with T-Shirt, and Stable Diffusion. On all 3 sets of experiments, SISS preserved the original model quality as shown in Figure 1. On CelebA-HQ with the objective of unlearning a celebrity face, SISS was Pareto-optimal with respect to the FID and SSCD similarity metric in Figure 2, cutting the latter by over half. The base model for MNIST with T-Shirt was trained on MNIST (Deng, 2012) augmented

with a specific T-shirt from Fashion-MNIST (Xiao et al., 2017). The objective was to unlearn the T-shirts, and SISS was again found to be Pareto-optimal with respect to the Inception Score and exact likelihood, increasing the latter by a factor of 8. Finally, on Stable Diffusion, we found SISS to successfully mitigate memorization on almost 90% of the prompts we tested.

## 2 RELATED WORK

**Machine Unlearning.**  Machine unlearning is the notoriously difficult problem of removing the influence of datapoints that models were previously trained on (Cao & Yang, 2015; Bourtoule et al., 2021; Shaik et al., 2024). Over the past years, it has received increased attention and relevance due to privacy regulation such as the EU's Right to be Forgotten (Ginart et al., 2019; Izzo et al., 2021; Golatkar et al., 2020; Tarun et al., 2023). The first wave of methods mostly approached classical machine learning methods like linear and logistic regression (Izzo et al., 2021), $k$-means clustering (Ginart et al., 2019), statistical query learning methods (Cao & Yang, 2015), Bayesian methods (Nguyen et al., 2020) or various types of supervised deep learning methods (Golatkar et al., 2020; Tarun et al., 2023). Some of these methods require modifications to the training procedure, e.g., training multiple models on distinct dataset shards (Bourtoule et al., 2021; Golatkar et al., 2024), whereas others can be applied purely in post-training such as NegGrad (Golatkar et al., 2020) and BlindSpot (Tarun et al., 2023). Recently, generative models have become a popular paradigm. While unlearning in this paradigm is less explored, there are early approaches looking at Generative Adversarial Networks (GANs) (Kong & Alfeld, 2023), language models (Liu et al., 2024; Yao et al., 2024) and on diffusion models via sharding (Golatkar et al., 2024).

**Memorization in Diffusion Models.**  Large-scale diffusion models trained on image generation have recently attracted the attention of copyright lawsuits since they are prone to memorizing training examples (Somepalli et al., 2023a; Carlini et al., 2023; Somepalli et al., 2023b; Webster, 2023). Somepalli et al. (2023a) showed that Stable Diffusion (Rombach et al., 2022) exhibits verbatim memorization for heavily duplicated training data examples. Webster (2023) classified different types of memorization, introducing types of partial memorization. Carlini et al. (2023) discusses various black-box extraction and membership inference attacks and demonstrates them successfully on Stable Diffusion. Most recently, mitigation strategies have been introduced, e.g., by manually modifying the text prompts (Somepalli et al., 2023b) or taking gradients steps in prompt space to minimize the magnitude of text-conditional noise predictions (Wen et al., 2024).

**Concept Unlearning in Diffusion Models.**  Recently, the problem of unlearning has become popular in the context of diffusion models, though almost exclusively in the form of concept unlearning: while the classical setting of machine unlearning deals with forgetting specific datapoints from the training set – which we call *data unlearning* for clarity – the setting of *concept unlearning* deals with forgetting higher level concepts in text-conditional models, e.g., nudity or painting styles (Shaik et al., 2024; Gandikota et al., 2023; Kong & Chaudhuri, 2024; Kumari et al., 2023; Zhang et al., 2024; Heng & Soh, 2023). Zhang et al. (2023) introduces Forget-Me-Not, a concept unlearning technique that minimizes the cross-attention map for an undesired prompt and also introduces ConceptBench as a benchmark. Gandikota et al. (2023) find an alternate approach to concept unlearning fine-tuning by fitting to noise targets that are biased away from the predicted noise with respect to an undesirable prompt. Similarly, Schramowski et al. (2023) also bias the noise away from an undesired prompt but do so only at inference time. UnlearnCanvas is a benchmark introduced by Zhang et al. (2024) to measure concept unlearning for artistic styles and objects. EraseDiff (Wu et al., 2024) discusses the data unlearning setting but only studies the settings of unlearning classes or concepts. Lastly, Li et al. (2024) indeed studies data unlearning but solely in the case of image-to-image models. To the authors' knowledge, the data unlearning setting remains a gap in the diffusion model literature.

## 3 PRELIMINARIES

We define the data unlearning problem as follows: given access to a training dataset $X = \{x_1, x_2, \ldots, x_n\}$ with $n$ datapoints and a diffusion model $\epsilon_\theta$ that was pretrained on $X$, our goal is to unlearn a $k$-element subset $A = \{a_1, a_2, \ldots, a_k\} \subset X$. We refer to $A$ as the *unlearning set*.

More specifically, we wish to efficiently unlearn $A$ through *deletion fine-tuning* which moves $\theta$ towards a set of parameters $\theta'$ so that $\epsilon_{\theta'}$ is no longer influenced by $A$. Ideally, $\epsilon_{\theta'}$ should behave as if it were trained from scratch on $X \setminus A$. In practice, however, retraining can be computationally infeasible. The key research question in data unlearning is identifying strategies for obtaining models that (1) preserve quality, (2) no longer generate $A$ unless generalizable from $X \setminus A$, and (3) are efficient.

Consider the standard DDPM forward noising process (Ho et al., 2020) where for a clean datapoint $x_0$, the noisy sample $x_t$ at time $t$ is given by

$$q(x_t|x_0) = \mathcal{N}(x_t; \gamma_t x_0, \sigma_t \mathbf{I}). \tag{1}$$

The parameters $\gamma_t$ and $\sigma_t$ are set by the variance schedule. A simple approach to data unlearning is by fine-tuning on $X \setminus A$ where the objective is to minimize the simplified evidence-based lower bound (ELBO):

$$L_{X \setminus A}(\theta) = \mathbb{E}_{p_{X \setminus A}(x)} \mathbb{E}_{q(x_t|x)} \|\epsilon - \epsilon_\theta(x_t, t)\|_2^2 \tag{2}$$

where $p_S$ refers to the discrete uniform distribution over any set $S$. We refer to this approach as *naive deletion* because it does not involve sampling from the unlearning set $A$. Another general-purpose machine unlearning approach is NegGrad (Golatkar et al., 2020) which performs gradient ascent on $A$, maximizing

$$L_A(\theta) = \mathbb{E}_{p_A(a)} \mathbb{E}_{q(a_t|a)} \|\epsilon - \epsilon_\theta(a_t, t)\|_2^2. \tag{3}$$

NegGrad is effective at unlearning $A$ but is unstable in that the predicted noise will grow in magnitude, and the model will eventually forget data from $X \setminus A$.

More recently, EraseDiff (Wu et al., 2024) unlearns by minimizing the objective

$$L_{X \setminus A}(\theta) + \lambda \mathbb{E}_{\epsilon_f \sim \mathcal{U}(\mathbf{0}, \mathbf{I}_d)} \mathbb{E}_{p_A(a)} \mathbb{E}_{q(a_t|a)} \|\epsilon_f - \epsilon_\theta(a_t, t)\|_2^2$$

through a Multi-Objective Optimization framework. Other state-of-the-art diffusion unlearning approaches such as SalUn (Fan et al., 2024) and Selective Amnesia (Heng & Soh, 2023) are designed only for conditional models and cannot be extended to the data unlearning setting. When unlearning a class $c$, they require either fitting to the predicted noise of a different class $c' \neq c$ or specifying a distribution $q(\mathbf{x} \mid \mathbf{c})$ to guide $\epsilon_\theta$ towards when conditioned on $c$, neither of which we assume access to.

## 4 PROPOSED METHOD: SUBTRACTED IMPORTANCE SAMPLED SCORES (SISS)

Assume the data unlearning setting from Section 3 with dataset $X$ of size $n$ and unlearning set $A$ of size $k$. The naive deletion loss from Eq. 2 can be split up as

$$
\begin{aligned}
L_{X \setminus A}(\theta) &= \mathbb{E}_{p_{X \setminus A}(x)} \mathbb{E}_{q(x_t|x)} \|\epsilon - \epsilon_\theta(x_t, t)\|_2^2 = \sum_{x \in X \setminus A} \frac{1}{n-k} \mathbb{E}_{q(x_t|x)} \|\epsilon - \epsilon_\theta(x_t, t)\|_2^2 \\
&= \sum_{x \in X} \frac{1}{n-k} \mathbb{E}_{q(x_t|x)} \|\epsilon - \epsilon_\theta(x_t, t)\|_2^2 - \sum_{a \in A} \frac{1}{n-k} \mathbb{E}_{q(a_t|a)} \|\epsilon - \epsilon_\theta(a_t, t)\|_2^2 \\
&= \frac{n}{n-k} \mathbb{E}_{p_X(x)} \mathbb{E}_{q(x_t|x)} \left\| \frac{x_t - \gamma_t x}{\sigma_t} - \epsilon_\theta(x_t, t) \right\|_2^2 \\
&\quad - \frac{k}{n-k} \mathbb{E}_{p_A(a)} \mathbb{E}_{q(a_t|a)} \left\| \frac{a_t - \gamma_t a}{\sigma_t} - \epsilon_\theta(a_t, t) \right\|_2^2.
\end{aligned}
\tag{4}
$$

By employing importance sampling (IS), we can bring the two terms from Eq. 4 together, requiring only one model forward pass as opposed to two forward passes through $\epsilon_\theta$ on both $x_t$ and $a_t$. IS restricts us to two choices: we either pick our noisy sample from $q(\cdot \mid x)$ or $q(\cdot \mid a)$. However, a

*defensive mixture distribution* (Hesterberg, 1995) parameterized by $\lambda$ allows us to weigh sampling between $q(\,\cdot\mid x)$ and $q(\,\cdot\mid a)$, giving us the following SISS loss function:

$$
\ell_\lambda(\theta) = \mathbb{E}_{p_X(x)}\mathbb{E}_{p_A(a)}\mathbb{E}_{q_\lambda(m_t|x,a)} \tag{5}
$$
$$
\left[\frac{n}{n-k}\frac{q\left(m_t|x\right)}{(1-\lambda)q\left(m_t|x\right)+\lambda q\left(m_t|a\right)}\left\|\frac{m_t-\gamma_t x}{\sigma_t}-\epsilon_\theta(m_t,t)\right\|_2^2\right.
$$
$$
\left.-\frac{k}{n-k}\frac{q\left(m_t|a\right)}{(1-\lambda)q\left(m_t|x\right)+\lambda q\left(m_t|a\right)}\left\|\frac{m_t-\gamma_t a}{\sigma_t}-\epsilon_\theta(m_t,t)\right\|_2^2\right]
$$

where $m_t$ is sampled from the mixture distribution $q_\lambda$ defined by a weighted average of the densities of $q(m_t|x)$ and $q(m_t|a)$

$$
q_\lambda\left(m_t|x,a\right) := (1-\lambda)q\left(m_t|x\right)+\lambda q\left(m_t|a\right). \tag{6}
$$

Employing IS and a defensive mixture distribution preserves the naive deletion loss. That is,

$$
\ell_\lambda(\theta) = L_{X\setminus A}(\theta) \forall \lambda \in [0,1]. \tag{7}
$$

We further prove that gradient estimators of the two loss functions used to update model parameters during deletion fine-tuning are also the same in expectation.

**Lemma 1.** *In expectation, gradient estimators of a SISS loss function $\ell_\lambda(\theta)$ and the naive deletion loss $L_{X\setminus A}(\theta)$ are the same.*

*Proof.* Follows from Eq. 7 and linearity of expectation. See Appendix A.2 for a complete proof.

Notice that the second term in Eq. 4 is the same as the NegGrad objective in Eq. 3 up to a constant. Hence, to boost unlearning on $A$, we increase its weight by a factor of $1 + s$ where $s > 0$ is a hyperparameter, referred to as the *superfactor*. The final weighted SISS loss $\ell_{s,\lambda}(\theta)$ can be written as

$$
\mathbb{E}_{p_X(x)}\mathbb{E}_{p_A(a)}\mathbb{E}_{q_\lambda(m_t|x,a)}\left[\frac{n}{n-k}\frac{q\left(m_t|x\right)}{(1-\lambda)q\left(m_t|x\right)+\lambda q\left(m_t|a\right)}\left\|\frac{m_t-\gamma_t x}{\sigma_t}-\epsilon_\theta(m_t,t)\right\|_2^2\right. \tag{8}
$$
$$
\left.-(1+s)\frac{k}{n-k}\frac{q\left(m_t|a\right)}{(1-\lambda)q\left(m_t|x\right)+\lambda q\left(m_t|a\right)}\left\|\frac{m_t-\gamma_t a}{\sigma_t}-\epsilon_\theta(m_t,t)\right\|_2^2\right].
$$

and is studied for stability and interpretability in Appendix A.1.

Despite Lemma 1, we find distinct SISS behavior for $\lambda = 0$ and $\lambda = 1$ that emulates naive deletion and NegGrad, respectively. We speculate that this discrepency is due to high gradient variance. Namely, for $\lambda = 0$, the SISS only selects noisy samples from $q(\,\cdot\,|x)$ that are high unlikely to come from $q(\,\cdot\,|a)$. As a result, the first term of the SISS loss dominates, resulting in naive deletion-like behavior. Similarly, for $\lambda = 1$, the second term of the SISS loss will dominate which matches NegGrad. Thus, in practice, we choose $\lambda = 0.5$ to ensure the beneficial properties of both naive deletion and NegGrad in SISS.

## 5 EXPERIMENTS

We evaluate our SISS method, its ablations, EraseDiff, NegGrad, and naive deletion through unlearning experiments on CelebA-HQ, MNIST T-Shirt, and Stable Diffusion. The SISS ablations are defined as follows:

**Setting $\lambda = 0$ and $\lambda = 1$.** Using $\lambda = 0$ or $\lambda = 1$ effectively disables the use of the mixture distribution and can be viewed as using only importance sampling.

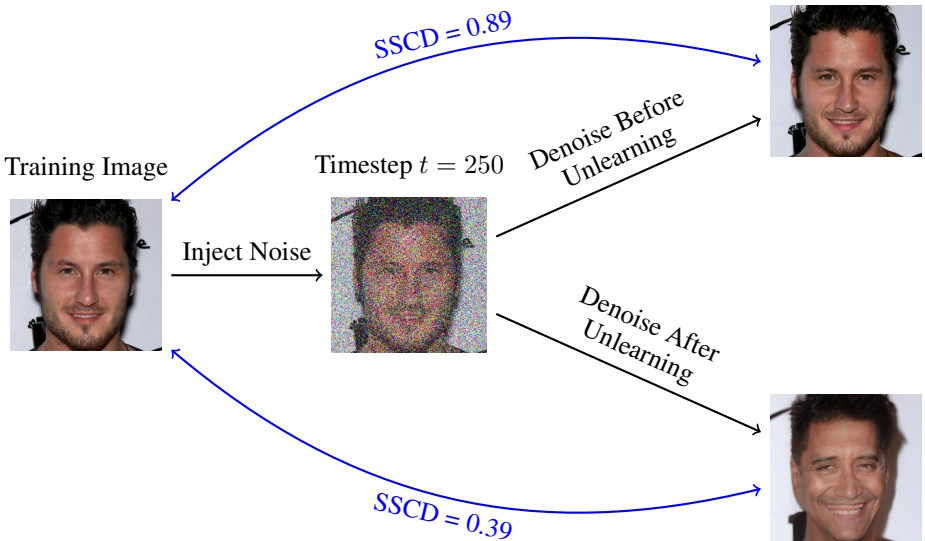

Figure 2: CelebA-HQ SSCD Metric Calculation. The process begins by taking the training face to be unlearned and injecting noise as part of the DDPM's forward noising process. Prior to unlearning, denoising the noise-injected face will result in a high similarity to the original training face. After unlearning, we desire for the denoised face to be significantly less similar to the training face.

**SISS (No IS).** The loss function defined in Eq. 4 after manipulating $L_{X \setminus A}(\theta)$ disables the use of importance sampling. Note, however, that it requires two forward passes through the denoising network $\epsilon_\theta$ and is thus *doubly more expensive in compute and memory*.

Our model quality metrics are standard and given by the Frechet Inception Distance (FID) (Heusel et al., 2017), Inception Score (IS) (Salimans et al., 2016), and CLIP-IQA (Wang et al., 2023). To evaluate the strength of unlearning, we employ the SSCD (Pizzi et al., 2022) to measure the similarity between celebrity faces before and after unlearning as in Figure 2. On MNIST T-Shirt and Stable Diffusion, we analyze the decay in the proportion of T-shirts and memorized images through sampling the model. Moreover, we also use the exact likelihood computation (Song et al., 2021b) that allows us to directly calculate the negative log likelihood (NLL) of the datapoint to unlearn. More details on the experimental setup and resources used are provided in Appendix B.

The objective across all 3 sets of experiments is to establish the Pareto frontier between model quality and unlearning strength.

To ensure stability of our SISS method, we adjust the superfactor $s$ in Eq. 8 so that the gradient norm of the second NegGrad term responsible for unlearning is fixed to around $10\%$ of the gradient norm of the first naive deletion term responsible for ensuring the model retains $X \setminus A$. This helps to control the second term's magnitude which can suffer from an exploding gradient. In Appendix A.3, we prove that for small step sizes, this gradient clipping adjustment reduces the naive deletion loss, thereby preserving model quality.

## 5.1 CELEBA-HQ

The CelebA-HQ dataset consists of 30000 high-quality celebrity faces (Karras et al., 2018). We use the unconditional DDPM trained by Ho et al. (2020) as our pretrained model. 6 celebrity faces were randomly selected to separately unlearn across all 7 methods. We found that injecting noise to timestep $t = 250$ on the celebrity face to be unlearned and denoising both before and after unlearning had a significant difference in similarity to the original face. Figure 2 illustrates this procedure and the quantification of similarity through the SSCD metric. The decrease in SSCD by over $50\%$ is observed visually in Figure 3 where SISS ($\lambda = 0.5$) and SISS (No IS) guide the model away from the celebrity to be unlearned. Guiding the model away from generating the celebrity face even with

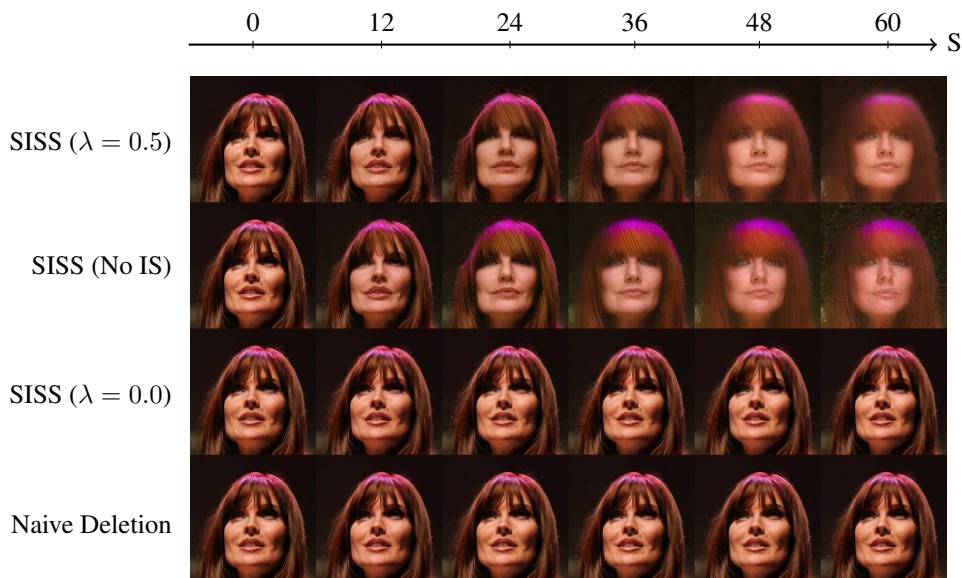

Figure 3: Visualization of celebrity unlearning over fine-tuning steps on quality-preserving methods. The images shown are made by applying noise to the original face and denoising as explained in Figure 2. Only SISS ($\lambda = 0.5$) and SISS (No IS) demonstrate the ability to guide the model away from generating the celebrity face.

Table 1: CelebA-HQ Unlearning Metrics. All methods were run for 60 fine-tuning steps on 6 separate celebrity faces. Methods in blue preserve the model quality, while the methods in red significantly decrease model quality. Only SISS ($\lambda = 0.5$) and SISS (No IS) are able to preserve model quality and unlearn the celebrity faces simultaneously relative to the pretrained model.

| Method | FID $\downarrow$ | NLL $\uparrow$ | SSCD $\downarrow$ |
|---|---|---|---|
| Pre-trained | 30.3 | 1.257 | 0.87 |
| Naive deletion | 19.6 | 1.240 | 0.87 |
| SISS ($\lambda = 0.0$) | 20.1 | 1.241 | 0.87 |
| **SISS ($\lambda = 0.5$)** | 25.1 | 1.442 | 0.36 |
| **SISS (No IS)** | 20.1 | 1.592 | 0.32 |
| EraseDiff | 117.8 | 4.445 | 0.19 |
| SISS ($\lambda = 1.0$) | 327.8 | 6.182 | 0.02 |
| NegGrad | 334.3 | 6.844 | 0.02 |

strong signal from timestep $t = 250$ indicates that the model is no longer incentivized to generate the face, especially at inference-time which starts from pure noise.

The SSCD, NLL, and FID unlearning and quality metrics averaged across faces are provided in Table 1. Figure 4a highlights the Pareto optimality of SISS ($\lambda = 0.5$) and SISS (No IS) along the FID and SSCD dimensions. All other methods either significantly increased the model FID or left the SSCD unaffected. In addition, we found that SISS ($\lambda = 0.5$) maintained high quality when unlearning 50 celebrity faces sequentially for 60 steps each with a final FID of 20.3, suggesting model stability over time.

## 5.2 MNIST T-Shirt

We train an unconditional DDPM on the MNIST dataset augmented with a T-shirt from Fashion-MNIST at a rate of $1\%$ (Xiao et al., 2017). This experiment serves as a toy setting of analyzing the unlearning behavior of a single data point. After training, we found that the model generated the T-shirt at a rate $p = 0.74\%$ with a $95\%$ confidence interval of $(0.68\%, 0.81\%)$. Table 2 highlights

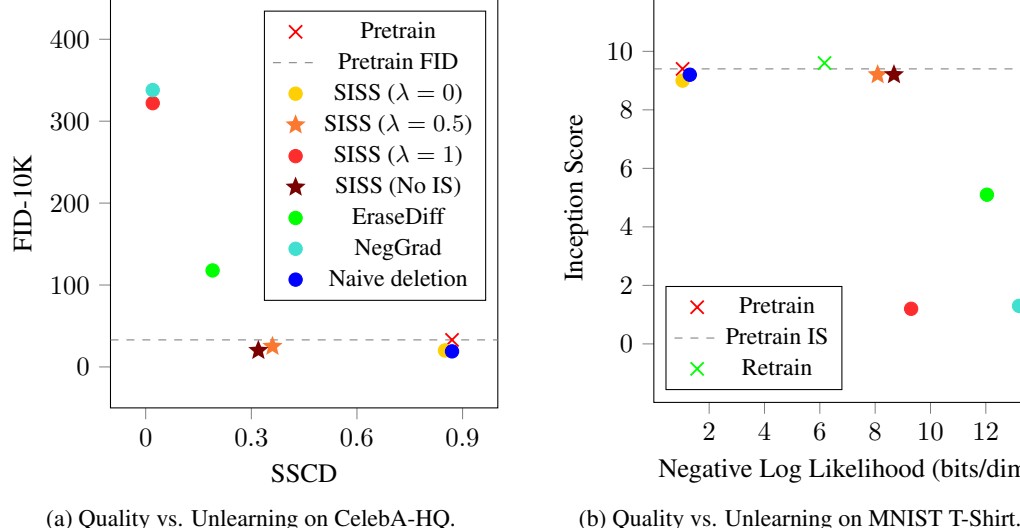

(a) Quality vs. Unlearning on CelebA-HQ.

(b) Quality vs. Unlearning on MNIST T-Shirt.

Figure 4: On both datasets, the only Pareto improvements over the pretrained model are given by SISS ($\lambda = 0.5$) and SISS (No IS). Remarkably, on MNIST T-Shirt, the two methods are Pareto improvements over the retrained model as well.

Table 2: MNIST T-Shirt Unlearning Metrics. $p$ represents the fraction of T-shirts observed from sampling 30720 images. Methods in blue preserve model quality, while methods in red significantly decrease model quality. All numbers are averaged across 5 seeds for each method. We illustrate the decay in $p$ as unlearning progresses for SISS ($\lambda = 0.5$).

| Method | Steps | IS ↑ | NLL ↑ | $p$ ↓ |
|---|---|---|---|---|
| Pre-trained | 117500 | 9.6 | 1.00 | 0.74% |
| Retrain | 117500 | 9.6 | 6.17 | 0% |
| Naive deletion | 300 | 9.5 | 1.19 | 0.04% |
| SISS ($\lambda = 0.0$) | 300 | 9.4 | 1.04 | 0.003% |
| **SISS ($\lambda = 0.5$)** | 300 | 9.2 | 8.09 | 0% |
| **SISS (No IS)** | 300 | 9.2 | 8.68 | 0% |
| EraseDiff | 100 | 5.1 | 12.04 | N/A |
| SISS ($\lambda = 1.0$) | 100 | 1.2 | 9.30 | N/A |
| NegGrad | 100 | 1.3 | 15.79 | N/A |

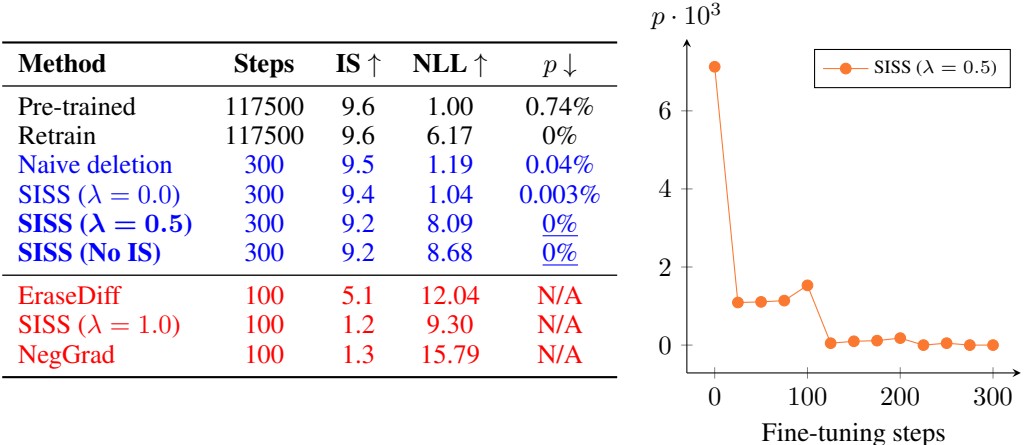

the rate of T-shirts after unlearning, showing that while naive deletion and SISS ($\lambda = 0$) significantly reduce the rate, only SISS ($\lambda = 0.5$) and SISS (No IS) are able to reach 0%.

Furthermore, Figure 4b highlights the Pareto optimality of SISS ($\lambda = 0.5$) and SISS (No IS) with respect to Inception Score and NLL even when including retraining. Much like the CelebA-HQ results, all other methods either significantly decreased the Inception Score or did not change the T-shirt's NLL except, of course, retraining.

## 5.3 STABLE DIFFUSION

We curate a set of 45 prompts that induce memorization on Stable Diffusion v1.4 drawn from Webster (2023). The objective is to unlearn the memorized training image from LAION corresponding to

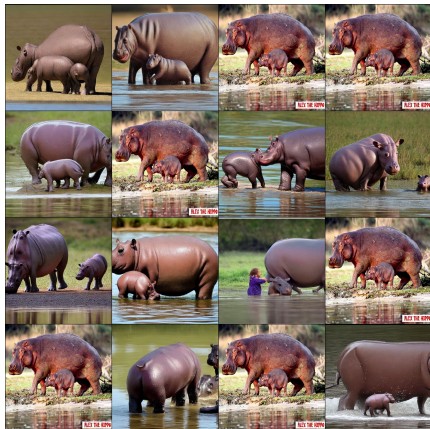
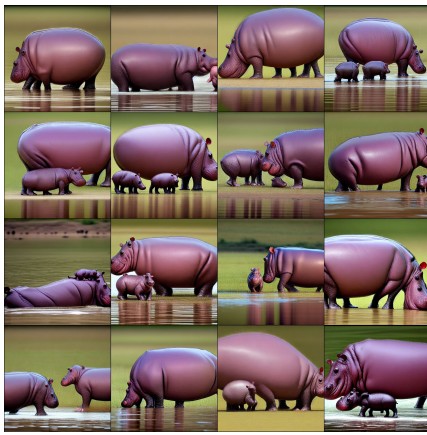

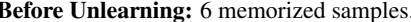

**Before Unlearning:** 6 memorized samples.          **After Unlearning:** 0 memorized samples.

Figure 5: Visualization of memorization mitigation on Stable Diffusion v1.4 using SISS ($\lambda = 0.5$). The number of memorized samples decreases from $6$ to $0$ on the partially-memorized prompt "Mothers influence on "her young hippo." Note the two apostrophes in red were purposefully inserted to turn the fully-memorized prompt into a partially-memorized prompt (see Section 5.3 for details).

each prompt. Stable Diffusion is a text-conditional model; however, by keeping the prompt fixed, it can be treated as an unconditional model which is where we perform unlearning.

SISS requires a set of training examples that include both unlearning set and non-unlearning set members. Applying it to memorization mitigation on Stable Diffusion requires addressing two fundamental issues: the lack of relevant training examples and the strong memorization of prompts.

**Lack of training examples.** For a given memorized prompt, there is only one corresponding training LAION image. However, our method relies on having access to a dataset $X \setminus A$. We instead synthetically generate this dataset by sampling 128 images for each prompt and using a $k$-means classifier for labelling each image as memorized ($A$) or not ($X \setminus A$).

**Strong memorization of prompts.** Many of the prompts sourced from Webster (2023) are *fully-memorized*, i.e. our synthetically-generated dataset from this prompt would exclusively contain examples of memorized images. Inspired by the prompt modification results of Somepalli et al. (2023b), we manually delete and add tokens to obtain a *partially-memorized* prompt that generates a greater frequency of non-memorized images on each of our $45$ prompts without fully mitigating memorization. The caption of Figure 5 shows an example of a partially-memorized prompt where two apostrophes were inserted to introduce sample diversity. For each prompt, we perform deletion fine-tuning on the synthetic dataset of its modified version.

We note that SISS ($\lambda = 0$) had numerical instability issues on Stable Diffusion because the NegGrad term is often very small, causing extremely large scaling factors. Thus, we excluded it from this experiment since it would be equivalent to naive deletion if scaling were disabled. Figure 6a shows that only SISS ($\lambda = 0.5$), SISS (No IS), and naive deletion are able to maintain high model quality as deletion fine-tuning occurs.

With respect to unlearning strength, Figure 6b illustrates that SISS ($\lambda = 0.5$) and SISS (No IS) are more successful in unlearning on the partially-memorized prompts than EraseDiff and naive deletion where success is the combination of reaching $0$ out of $16$ memorized samples and maintaining a CLIP-IQA of at least $0.35$ throughout deletion fine-tuning. In addition, the two SISS methods exhibit better unlearning generalization to the fully-memorized prompts, suggesting that the model updates done with the partially-memorized prompt extend to the fully-memorized prompt in latent space.

## 6 CONCLUSION

Prior methods in diffusion unlearning have been focused on class and concept unlearning. We introduce SISS, a novel method for data unlearning in diffusion models that utilizes importance

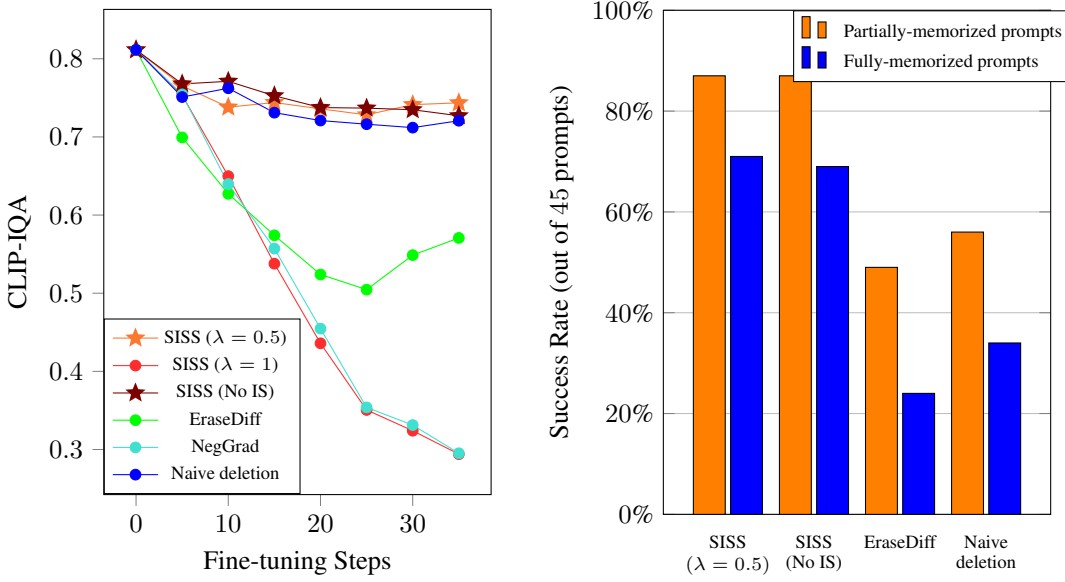

(a) Quality as unlearning progresses.

(b) Success rate among quality-preserving methods.

Figure 6: Stable Diffusion Quality and Unlearning Results. SISS ($\lambda = 0.5$) and SISS (No IS) preserve model quality as strongly as naive deletion. EraseDiff has a moderately negative impact on quality, while the other methods significantly degrade quality. Naive deletion and EraseDiff have noticably poorer success when compared to our SISS methods and do not generalize well to the fully-memorized prompts. Success is defined as a run having no memorized image outputted in 16 samples at the end, and the average CLIP-IQA score being at least $0.35$ throughout the run.

sampling for computational efficiency. Our method is able to effectively unlearn training datapoints while maintaining model quality. It exhibits Pareto optimality on multiple datasets across the quality and unlearning dimensions as well as strong memorization mitigation performance on text-conditional models such as Stable Diffusion. In the future, we hope to analyze the unlearning generalization across prompts in more detail and find a way around the "prompt modification" step that is hard to automate. Additional future directions include analyzing data unlearning on other data modalities such as audio and video. To finish, we remark that while our method may be successful in unlearning, it may not be enough legally for copyrighted data since that data is a part of the unlearning process itself.

ACKNOWLEDGMENTS

We would like to thank Dongjun Kim for his detailed review of this manuscript and insightful discussions on the behavior of SISS.

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

# A SISS MATH

## A.1 STABILITY ANALYSIS AND INTERPRETATION OF SISS

Recall that the weighted loss $\ell_{s,\lambda}(\theta)$ is

$$
\mathbb{E}_{p_X(x)}\mathbb{E}_{p_A(a)}\mathbb{E}_{q_\lambda(m_t|x,a)}\left[\frac{n}{n-k}\frac{q(m_t|x)}{(1-\lambda)q(m_t|x)+\lambda q(m_t|a)}\left\|\frac{m_t-\gamma_t x}{\sigma_t}-\epsilon_\theta(m_t,t)\right\|_2^2\right.
$$

$$
\left. - (1+s)\frac{k}{n-k}\frac{q(m_t|a)}{(1-\lambda)q(m_t|x)+\lambda q(m_t|a)}\left\|\frac{m_t-\gamma_t a}{\sigma_t}-\epsilon_\theta(m_t,t)\right\|_2^2\right].
$$

An advantage of sampling from the defensive mixture distribution $q_\lambda$ is that the importance weights of the noise norms are bounded for $0 < \lambda < 1$

$$0 \le \frac{q\left(m_t|x\right)}{(1-\lambda)q\left(m_t|x\right) + \lambda q\left(m_t|a\right)} \le \frac{1}{1-\lambda} \tag{9}$$

$$0 \le \frac{q\left(m_t|a\right)}{(1-\lambda)q\left(m_t|x\right) + \lambda q\left(m_t|a\right)} \le \frac{1}{\lambda}, \tag{10}$$

ensuring greater numerical stability during deletion fine-tuning. The choice of writing the superfactor as $1 + s$ allows us to rewrite the outermost expectation to sample from $p_{X \setminus A}$ instead of $p_X$:

$$\ell_{s,\lambda}(\theta) = \sum_{x \in X} \frac{1}{n-k} \mathbb{E}_{q(x_t|x)} \left\| \epsilon - \epsilon_\theta(x_t, t) \right\|_2^2 \tag{11}$$

$$- (1+s) \sum_{a \in A} \frac{1}{n-k} \mathbb{E}_{q(a_t|a)} \left\| \epsilon - \epsilon_\theta(a_t, t) \right\|_2^2$$

$$= \sum_{x \in X \setminus A} \frac{1}{n-k} \mathbb{E}_{q(x_t|x)} \left\| \epsilon - \epsilon_\theta(x_t, t) \right\|_2^2 \tag{12}$$

$$- s \sum_{a \in A} \frac{1}{n-k} \mathbb{E}_{q(a_t|a)} \left\| \epsilon - \epsilon_\theta(a_t, t) \right\|_2^2$$

$$= \mathbb{E}_{p_{X \setminus A}(x)} \mathbb{E}_{q(x_t|x)} \left\| \epsilon - \epsilon_\theta(x_t, t) \right\|_2^2 \tag{13}$$

$$- s \frac{k}{n-k} \mathbb{E}_{p_A(a)} \mathbb{E}_{q(a_t|a)} \left\| \epsilon - \epsilon_\theta(a_t, t) \right\|_2^2$$

$$= \mathbb{E}_{p_{X \setminus A}(x)} \mathbb{E}_{q(x_t|x)} \left\| \frac{x_t - \gamma_t x}{\sigma_t} - \epsilon_\theta(x_t, t) \right\|_2^2 \tag{14}$$

$$- s \frac{k}{n-k} \mathbb{E}_{p_A(a)} \mathbb{E}_{q(a_t|a)} \left\| \frac{a_t - \gamma_t a}{\sigma_t} - \epsilon_\theta(a_t, t) \right\|_2^2$$

$$= \mathbb{E}_{p_{X \setminus A}(x)} \mathbb{E}_{p_A(a)} \mathbb{E}_{q_\lambda(m_t|x,a)} \tag{15}$$

$$\left[ \frac{q\left(m_t|x\right)}{(1-\lambda)q\left(m_t|x\right) + \lambda q\left(m_t|a\right)} \left\| \frac{m_t - \gamma_t x}{\sigma_t} - \epsilon_\theta(m_t, t) \right\|_2^2 \right.$$

$$\left. - s \frac{k}{n-k} \frac{q\left(m_t|a\right)}{(1-\lambda)q\left(m_t|x\right) + \lambda q\left(m_t|a\right)} \left\| \frac{m_t - \gamma_t a}{\sigma_t} - \epsilon_\theta(m_t, t) \right\|_2^2 \right].$$

From Eq. 12, we see that the weighted loss $\ell_{s,\lambda}(\theta)$ has two separate terms: the naive deletion loss $L_{X \setminus A}(\theta)$ and a term proportional to the NegGrad loss that discourages the generation of $A$'s members. Thus, Eq. 12 is equivalent to

$$L_{X \setminus A}(\theta) - s \frac{k}{n-k} L_A(\theta)$$

where the superfactor $s$ controls the weight of the NegGrad term $L_A$. As a result, we can directly view $\ell_{s,\lambda}$ as interpolating between naive deletion and NegGrad with $s$ controlling the strength of NegGrad.

Notice that if $t$ is small then $q(m_t|x) \gg q(m_t|a)$ if $m_t$ is sampled from $q(\cdot|x)$ and $q(m_t|x) \ll q(m_t|a)$ if $m_t$ is sampled from $q(\cdot|a)$. If $\lambda = 0$, we know $q(m_t|x) \gg q(m_t|a)$, making the importance ratio of the first term in the SISS loss equal to $1$, and the second importance ratio equal to $0$. Hence, SISS with $\lambda = 0$ will be equivalent to naive deletion. Similarly, if $\lambda = 1$, $q(m_t|x) \ll q(m_t|a)$ and the second importance ratio will dominate, which is equivalent to NegGrad.

## A.2 PROOF OF LEMMA 1

**Lemma 1 Restated.** Gradient estimators of $\ell_\lambda(\theta)$ and $L_{X \setminus A}(\theta)$ are the same in expectation. That is, in expectation, Monte Carlo estimates of

$$\nabla_\theta \ell_\lambda(\theta) = \mathbb{E}_{p_X(x)} \mathbb{E}_{p_A(a)} \mathbb{E}_{q_\lambda(m_t|x,a)} \tag{16}$$
$$\left[ \nabla_\theta \left( \frac{n}{n-k} \frac{q\left(m_t|x\right)}{(1-\lambda)q\left(m_t|x\right) + \lambda q\left(m_t|a\right)} \left\| \frac{m_t - \gamma_t x}{\sigma_t} - \epsilon_\theta(m_t, t) \right\|_2^2 \right. \right.$$
$$\left. \left. - \frac{k}{n-k} \frac{q\left(m_t|a\right)}{(1-\lambda)q\left(m_t|x\right) + \lambda q\left(m_t|a\right)} \left\| \frac{m_t - \gamma_t a}{\sigma_t} - \epsilon_\theta(m_t, t) \right\|_2^2 \right) \right]$$

and Monte Carlo estimates of

$$\nabla_\theta L_{X \setminus A}(\theta) = \mathbb{E}_{p_{X \setminus A}(x)} \mathbb{E}_{q(x_t|x)} \left[ \nabla_\theta \left\| \epsilon - \epsilon_\theta(x_t, t) \right\|_2^2 \right] \tag{17}$$

are equal.

*Proof.* Note that Eqs. 16 and 17 are direct consequences of the linearity of expectation which allows us to take the gradient with respect to $\theta$ inside the expectation operations. The equivalence of the two loss functions (Eq. 7) implies that

$$\nabla_\theta \ell_\lambda(\theta) = \nabla_\theta L_{X \setminus A}(\theta). \tag{18}$$

When combined with Eqs. 16 and 17, we see that the Monte Carlo gradient estimators are the same in expectation. □

## A.3 GRADIENT CLIPPING

Recall that we adjust the superfactor $s$ in Eq. 8 so that the gradient norm of the second NegGrad term responsible for unlearning is fixed to around $10\%$ of the gradient norm of the first naive deletion term responsible for ensuring the model retains $X \setminus A$. We show that for small step sizes this gradient clipping adjustment reduces the naive deletion loss, thus preserving the quality of the model. To prove this, we start with a variant of the classic descent lemma.

**Descent lemma under small perturbations.** Suppose the function $f : \mathbb{R}^n \to \mathbb{R}$ is differentiable, and that its gradient is Lipschitz continuous with constant $L > 0$, i.e., we have that $\|\nabla f(x) - \nabla f(y)\|_2 \le L\|x - y\|_2$ for any $x, y$. Then, one step of gradient descent from $x \in \mathbb{R}^n$ with step size $t$ and perturbed update

$$x' = x - t(\nabla f(x) + v)$$

satisifes the improvement bound

$$f(x') \le f(x) - (1 - p - \epsilon)t\|\nabla f(x)\|^2.$$

We assume

$$\epsilon = \frac{1}{2}Lt + Ltp + \frac{1}{2}Ltp^2,$$

and $v \in \mathbb{R}^n$ is an arbitrary vector satisfying

$$\|v\| \le p\|\nabla f(x)\|$$

for a proportion $p$.

*Proof.* $\nabla f$ being $L$-Lipschitz implies

$$f(x') \le f(x) + \nabla f(x)^\top (x' - x) + \frac{L}{2}\|x' - x\|^2.$$

Plugging in our update equation and repeatedly applying Cauchy-Schwarz gives

$$f(x') \le f(x) + \nabla f(x)^\top (x' - x) + \frac{L}{2}\|x' - x\|^2$$
$$= f(x) + \nabla f(x)^\top (-t\nabla f(x) - tv) + \frac{L}{2}\| - t\nabla f(x) - tv\|^2$$

$$= f(x) - t\|\nabla f(x)\|^2 - t\nabla f(x)^\top v + \frac{Lt^2}{2} \langle \nabla f(x) + v, \nabla f(x) + v \rangle$$

$$= f(x) - t\|\nabla f(x)\|^2 - t\nabla f(x)^\top v + \frac{Lt^2}{2} \left( \|\nabla f(x)\|^2 + 2\nabla f(x)^\top v + \|v\|^2 \right)$$

$$\leq f(x) - t\|\nabla f(x)\|^2 + t\|\nabla f(x)\|\|v\| + \frac{Lt^2}{2} \left( \|\nabla f(x)\|^2 + 2\|\nabla f(x)\|\|v\| + \|v\|^2 \right)$$

$$\leq f(x) - t\|\nabla f(x)\|^2 + tp\|\nabla f(x)\|^2 + \frac{Lt^2}{2} \left( \|\nabla f(x)\|^2 + 2p\|\nabla f(x)\|^2 + p^2\|\nabla f(x)\|^2 \right)$$

$$= f(x) - t\left( 1 - p - \frac{Lt}{2} - Ltp - \frac{Ltp^2}{2} \right) \|\nabla f(x)\|^2.$$

Hence, setting

$$\epsilon = \frac{1}{2}Lt + Ltp + \frac{1}{2}Ltp^2$$

gives the desired result where we note that $t \to 0$ implies $\epsilon \to 0$. Thus, $\epsilon$ can be made arbitrary small by selecting a small step size $t$. $\quad\square$

Notice that the classic descent lemma corresponds to the special case of $p = 0$. Practically, as long as $p < 1$, we can choose $t$ small so that we are guaranteed improvement on each step of gradient descent. Standard results show that this implies gradient descent will eventually converge to a point where $\|\nabla f(x)\| < \delta$ in $\mathcal{O}(\frac{1}{\delta})$ iterations.

In the data unlearning context, set $f$ to be the naive deletion objective and pick $v$ to be a rescaled version of the NegGrad objective's gradient. Our perturbed descent lemma shows that for appropriate step size we will be no worse off in naive deletion performance, implying that model quality will be preserved. It is not theoretically clear that choosing $v$ to be NegGrad's gradient will result in unlearning. However, our results in Section 5 found this to be empirically true.

## B  EXPERIMENTAL SETUP

All diffusion models were trained and fine-tuned using the Hugging Face `diffusers` package along with the Adam optimizer (Kingma & Ba, 2015). YAML configuration files with all run settings can be found in the `config/` directory of our codebase.

The CelebA-HQ experiments used a pretrained checkpoint from Ho et al. (2020) hosted at `https://huggingface.co/google/ddpm-celebahq-256`. Our pretrain and retrain unconditional MNIST T-Shirt DDPMs were trained for 250 epochs with a batch size of 128 images and a learning rate of $1e - 4$ with cosine decay. Both models used the same DDPM sampler at inference with 50 backwards steps. For the Stable Diffusion experiments, we used version 1.4 hosted at `https://huggingface.co/CompVis/stable-diffusion-v1-4` as our pretrained checkpoint with 50-step DDIM as the sampler (Song et al., 2021a). The models for all 3 sets of experiments use a U-Net backbone.

Deletion fine-tuning experiments were run starting from the EMA versions of the trained MNIST T-Shirt DDPMs as well as the pre-trained Stable Diffusion model. For MNIST T-Shirt, the same hyperparameters were kept from pretraining to run fine-tuning. In the case of CelebA-HQ and Stable Diffusion, we did not perform the pretraining and chose a batch size of 64 and 16 images with a learning rate of $5e - 6$ and $1e - 5$, respectively.

While most individual experiments were not very computationally expensive (roughly half an hour on average), sweeping across all different baselines and and mixture parameters $\lambda$ totaled to over 500 runs. To streamline this process, a cluster of 8 NVIDIA H100 GPUs were used to execute large numbers of runs in parallel. In addition, an g5.xlarge instance with an NVIDIA A10G GPU on AWS, a personal home computer with an NVIDIA RTX 3090, and a cluster of 3 NVIDIA A4000 GPUs were the primary code development environments.

