# OpenReview forum: "Data Unlearning in Diffusion Models"
_ICLR.cc/2025/Conference — ICLR 2025 Poster_

### Official Review · Reviewer_JqNx · 2024-10-31

**Soundness:** 3
**Presentation:** 3
**Contribution:** 2
**Rating:** 3
**Confidence:** 3

**Summary:**

The paper present a new data unlearning objective, Subtracted Importance Sampled Score (SISS). To efficiently compute SISS, the paper devices a new objective using importance sampling and defensive mixture distribution. Then the authors provide some empirical results to prove the effectiveness of the proposed objective.

**Strengths:**

- The paper is mostly well written.
- The paper conducts various experiments, including some ablations, to prove the effectiveness of the proposed method.

**Weaknesses:**

- The authors did not justify why *"involving sampling from the unlearning set A"* is beneficial, which seems to be the most important motivation of the proposed work.
- The choice of defensive mixture distribution $q_\lambda(m_t | x, a)$ seems not suited for diffusion training objective. According to the definition, it is a mixture of two normal distributions, and I believe sampling $m_t$ from such distribution will generate some sort of interpolated version of two noisy images. I'm not sure if this is a good noisy sample for the denoising objective of diffusion model. The actual sampling example from Figure 2 hints an overlay of two different images, which might be the innate problem caused by the objective.
- The need for IS is not clear. SISS (no IS) shows superior quality over SISS (with IS), and since the computational overhead only occurs during training, this, in my opinion, might not be a big problem. Also if involving samples from A is the core motive, one might consider objectives like $(1 - \delta_{x\in A})L(\theta, x) - \lambda (\delta_{x\in A})L(\theta, x)$, where $\lambda$ is the magnitude hyperparameters. This requires only one forward pass, and seems to resemble the spirit of Equation (4).

**Questions:**

- Why is involving samples from A important for unlearning?
- How do you calculate IS ratio?
- For CelebA example, why was **6** chosen as the number of samples to remove?
- Is the objective suggested in **Weaknesses** inferior to proposed SISS (empirically)?
- (Not so related to this work) Most samples in Figure 5 seems cartoonish. In fact, the only realistic hippo images are memorized ones. According to [1],  memorization can happen at concept-level, and [2] conjectures that memorization can be induced by lack of samples from specific concepts. Perhaps only realistic hippo related to the prompt in training set is the memorized one, and rather than unlearning, adding more realistic images for given prompt might solve the memorization problem.


[1] Somepalli et al. "Diffusion Art or Digital Forgery? Investigating Data Replication in Diffusion Models", CVPR 2023

[2] Yoon et al. "Diffusion Probabilistic Models Generalize when They Fail to Memorize", ICML 2023 Workshop on Structured Probabilistic Inference & Generative Modeling, 2023

---

> ### Author Response · Authors · 2024-12-01
>
> We appreciate the reviewer's comment stating our paper is “mostly well-written.”
>
> **Weaknesses**:
> 1. The claim that sampling from the unlearning set $A$ is beneficial for our loss is empirically justified by our ablations over $\lambda$:
>     - $\lambda=0$: Only sample from $X\setminus A$
>     - $\lambda=1$: Only sample from $A$
>     - $\lambda=0.5$: Sample from both $X\setminus A$ and $A$
>
>     All our experiments in the paper (e.g., Tables 1 and 2) show that $\lambda=0.5$ has significantly better unlearning performance than $\lambda=0$ which shows why sampling from the unlearning set $A$ in the loss function is crucial for our method.
>
> 2. It is not true that sampling from a mixture of two normal distributions generates an interpolated version of two noisy images. You might have been thinking of the linear combination of samples from two normal distributions (which would indeed be an interpolation and would cause strange effects). Our objective can intuitively be thought of: With $\lambda=0.5$ probability, we sample from a normal centered at $x\in X\setminus A$, with $1-\lambda=0.5$ probability, we sample normal centered at $a\in A$.
>
> 3. The reason why we present both variants SISS (no IS) and SISS (with IS) of our loss function is because SISS (no IS) might be the preferred choice for many readers. Indeed, the computational overhead only occurs at training and is negligible in many cases, so SISS (no IS) might be preferred because of its simplicity. However, SISS (with IS) uses half as many forward passes per step and achieves similar results. In particular, it uses half as much GPU memory which is relevant for many practitioners. Moreover, we found the importance sampling variant interesting because of its theoretical guarantees in being unbiased.
>
>     We also appreciate the suggested alternative loss function and would be glad to explore it in future work. The proposed loss function would be equal to SISS (with IS), wherever one of the importance weights is zero. However, wherever both of the importance weights are non-zero, SISS (with IS) has effectively 2x the batch size compared to the proposed loss due to incorporating both naive deletion and NegGrad terms. Empirically, for higher timesteps the importance weights are non-zero (see MNIST IS ratios at https://drive.google.com/file/d/1SyqVq1kLMMNdrtbcfH1M1lfqErvioOWm/view?usp=sharing) and for lower timesteps one weight is zero. Intuitively, the higher timesteps are important because they are where the diffusion process decides whether to collapse into the memorized image.
>
> **Questions**:
> 1.  See item 1 in weaknesses.
> 2.  Recall that the IS ratios in the SISS loss (Eq. 8 of our paper) are given by:
> $$\frac{q\left(m_t| x\right)}{(1-\lambda) q\left(m_t| x\right)+\lambda q\left(m_t| a\right)}= \frac{1}{(1 - \lambda) + \lambda \frac{q(m_t | a)}{q(m_t | x)}}$$
> and
> $$\frac{q\left(m_t| a\right)}{(1-\lambda) q\left(m_t| x\right)+\lambda q\left(m_t| a\right)}= \frac{1}{(1 - \lambda) \frac{q(m_t | x)}{q(m_t | a)} + \lambda}.$$
>
> By definition of DDPM, the densities of the noise-injecting Gaussians are given by:
> $$q(m_t | x) = \mathcal{N}(m_t; \gamma_t x, \sigma_t^2 I) = \frac{1}{(2\pi \sigma_t^2)^{d/2}} \exp\left(-\frac{1}{2\sigma_t^2} \|m_t - \gamma_t x\|_2^2\right),$$
> $$q(m_t | a) = \mathcal{N}(m_t; \gamma_t a, \sigma_t^2 I) = \frac{1}{(2\pi \sigma_t^2)^{d/2}} \exp\left(-\frac{1}{2\sigma_t^2} \|m_t - \gamma_t a\|_2^2\right),$$
>
> Notice that in order to compute the importance ratios, we need to find the ratio of the density of the above Gaussians which is given by:
> $$\frac{q(m_t | a)}{q(m_t | x)} = \exp\left(-\frac{1}{2\sigma_t^2} \|m_t - \gamma_t a\|_2^2 + \frac{1}{2\sigma_t^2} \|m_t - \gamma_t x\|_2^2\right),$$
>
> $$\frac{q(m_t | x)}{q(m_t | a)} = \exp\left(-\frac{1}{2\sigma_t^2} \|m_t - \gamma_t x\|_2^2 + \frac{1}{2\sigma_t^2} \|m_t - \gamma_t a\|_2^2\right).$$
>
> 3. For each CelebA experiment we selected one celebrity face datapoint to unlearn ($|A|=1$). To reduce variance, we ran the experiment $n=6$ times with randomly chosen data points and reported average metrics in Table 1 and Figure 4a. We chose the number $n=6$ because that was the maximum that fit in our computational budget.
>
> 4. See item 3 in weaknesses.

---

### Official Review · Reviewer_unz3 · 2024-11-03

**Soundness:** 3
**Presentation:** 3
**Contribution:** 3
**Rating:** 8
**Confidence:** 2

**Summary:**

The authors propose a new method to address data unlearning in diffusion models. If a diffusion model is trained on a set X, and the goal is to remove a set A, then the resulting diffusion model should not produce any element of A, unless it can be created from the set X \ A. The authors state that the data unlearning process should maintain model quality and be efficient (more efficient that retraining the model from scratch with X \ A).

For data unlearning, the authors introduce a new family of loss functions, Subtracted Importance Sampled Scores (SISS), which balances fine-tuning with the data remaining after subtracting A from X (naive deletion) and gradient ascent on the set A (NegGrad).

The authors show that two variations of their method lead to the best results: 1) SISS (lambda = 0.5), where naive deletion and NegGrad are balanced by a hyperparameter lambda and importance sampling is used to improve efficiency, and 2) SISS (no IS), where lambda is not needed and importance sampling is not used.

**Strengths:**

The authors introduce a new method for data unlearning in diffusion models, SSIS. The paper is well written, clearly defining the problem and motivating the need for this approach. Furthermore, the results with their approach are very promising. The authors use standard benchmarks and evaluation metrics, which support that their method is effective and efficient. Their experiments are well designed. Overall, this is a strong paper that makes a valuable contribution to the field.

**Weaknesses:**

1. A brief discussion on the limitations of generating synthetic data and using k-means clustering for Stable Diffusion would strengthen the analysis. Additionally, including a qualitative analysis of similarity, rather than relying solely on quantitative measures (k-means clustering) would provide additional support your method.
2. A discussion on why your method's performance, for the given metrics, surpasses the gold standard (retraining) with MNIST T-Shirt (Figure 4b) would be helpful.

**Questions:**

Please address the weaknesses.

---

> ### Author Response · Authors · 2024-12-01
>
> We appreciate the reviewer’s assessment that our paper is “well written” and “makes a valuable contribution to the field.”
>
> **Weaknesses:**
> 1. One limitation of generating synthetic data for SD lies with how SISS was derived based on unlearning training samples. Our synthetic examples were not used to train SD but were empirically sufficient for unlearning purposes as seen in Figure 6 of the paper. However, the primary limitation was in the manual construction of the modified prompts detailed in Figure 5. This process required manual trial and error of inserting and deleting tokens to obtain greater sample diversity.
>
> 2. The interpretation of SISS being an improvement over retraining in Figure 4(b) means it is less likely to generate a T-Shirt due to the way our method specifically targets unlearning it. While retraining doesn’t produce any T-Shirts anyway, there is nevertheless some likelihood, however small it may be, of generating a T-Shirt. With SISS, this likelihood is even smaller.

---

### Official Review · Reviewer_xvNv · 2024-11-04

**Soundness:** 2
**Presentation:** 3
**Contribution:** 3
**Rating:** 6
**Confidence:** 3

**Summary:**

This paper proposes a new family of loss objectives to perform data unlearning in diffusion models. Based on former work of defensive mixture distributions and by employing importance sampling, these loss objectives can be tuned to unlearn sub-datasets with less degraded quality in retained images.

**Strengths:**

1. The paper is well-written and easy to follow.
2. The proposed method is simple, and only based on light mathematical derivations.
3. The numerical performance is promising, showing significant improvements over reported baselines.

**Weaknesses:**

1. Although the experiments in this work are promising, there are insufficient theoretical justifications in some sense. I understand that this may be beyond the current scope, but this work can be strengthened if discussing more about working principles. For example, is it possible to introduce (mathematical) tools of traditional machine unlearning (for classical supervised machine learning) for more rigorous demonstrations? What are the main potential difficulties and differences when it comes to the generative setting?

2. The ablation studies can be more comprehensive (see details in the "Questions" section).

**Questions:**

1. Please kindly provide more details to the questions raised in the "Weaknesses" section.

2. Ablations:
- It seems that only $\lambda=0, 1$ are tested. What about other values of $\lambda$?
- In addition, how does the "superfactor" $s$ in Eq. (8) effect the unlearning performance numerically?

3. Experimental illustrations:
- It is shown that the method "SISS (No IS)" appears similar (even mostly better) performance with SISS ($\lambda=0.5$) across all experiments. Does this imply that it is unnecessary to introduce defensive mixture distributions?
- Also, there are missing 0/1 values of $\lambda$ to be tested (see Fig. 3 (no $\lambda=1$) and Fig. 6(a) (no $\lambda=0$)).

4. Can authors explain in more details on how to avoid training from scratch using the proposed loss objectives (since Eq. (4) also involves the expectation w.r.t. $p_X$)?

**Details Of Ethics Concerns:**

There are no ethics concerns.

---

> ### Author Response · Authors · 2024-12-01
>
> We appreciate the reviewer’s assessment that our paper is “well-written” and “makes a valuable contribution to the field.”
>
> **Weaknesses:**
> 1. For greater theoretical justification, we have proven a new result that shows for small learning rates, all SISS variants (with IS and no IS) must improve wrt. naive deletion loss. As a result, this provides a guarantee that model quality will not degrade. You can find a PDF with details of this lemma at https://drive.google.com/file/d/1B1rDuibatkhVp33ducUGKeSi9m69-lSp/view?usp=sharing. The intuition is that the gradient clipping we use makes it so that the naive deletion gradient dominates the NegGrad gradient, so the majority of gradient updates goes towards preserving model quality (i.e., naive deletion). The lemma does not provide guarantees around whether unlearning will occur, but our empirical results indicate that unlearning does occur.
>
> 2. Addressed below in questions.
>
> **Questions:**
>
> 2. Here are additional results for $\lambda=0.25, 0.75$:
> ### CelebA-HQ (Table 1 in Paper)
>
> | Method              | FID  | NLL   | SSCD |
> |---------------------|-------|-------|------|
> | SISS ($λ = 0.25$)     | 24.6  | 1.433 | 0.39 |
> | SISS ($λ = 0.5$)      | 25.1  | 1.442 | 0.36 |
> | SISS ($λ = 0.75$)     | 27.7  | 1.454 | 0.36 |
>
> ### MNIST T-Shirt (Table 2 in Paper)
>
> | Method              | Steps | IS   | NLL   | p   |
> |---------------------|-------|-------|-------|-----|
> | SISS ($λ = 0.25$)     | 300   | 9.3   | 7.98  | 0%  |
> | SISS ($λ = 0.5$)      | 300   | 9.2   | 8.09  | 0%  |
> | SISS ($λ = 0.75$)     | 300   | 9.1   | 8.18  | 0%  |
>
>
> As long as $\lambda$ is not too far from $0.5$, results will be similar. Generally, the objective of IS is to reduce estimator variance. $\lambda$, in particular, controls the tradeoff in variance between naive deletion and NegGrad. If $\lambda$ is close to $1$, then we will have a good estimate for NegGrad but a poor one for naive deletion and vice versa for $\lambda\approx 0$.
>
> The superfactor has a significant impact on quality degradation. With SISS, the superfactor is auto-tuned to keep the NegGrad gradient norm at a fixed fraction of the naive deletion gradient norm. If this clipping-based setting of the superfactor is not done, then model quality significantly degrades because of NegGrad’s exploding gradient, rendering unlearning results meaningless.
>
> 3. The reason why we present both variants SISS (no IS) and SISS ($\lambda=0.5$) of our loss function is because SISS (no IS) might be the preferred choice for many readers. The computational overhead of SISS (no IS) is negligible in many cases and might be preferred because of its simplicity. However, SISS ($\lambda=0.5$) uses half as many forward passes per step and achieves similar results. In particular, it uses half as much GPU memory which is relevant for many practitioners. Moreover, we found the importance sampling variant interesting because of its theoretical guarantees in being unbiased.
>
>     Regarding $\lambda=1$ missing from Figure 3, it was not included because of the quality degradation quantitatively shown in Table 1. For $\lambda=0$ missing in Figure 6, we briefly mentioned why in lines 476-478 of the paper. Namely, SISS ($\lambda=0$) had numerical instability issues from very small NegGrad terms. Removing these NegGrad terms would result in identical behavior to naive deletion.
>
>
> 4. There is no need to train from scratch with any of our proposed loss objectives. We do assume access to the training data $X$ and unlearning set $A$ in order to sample from them in Eq. (4). However, all our experiments start from a pretrained model checkpoint and fine-tune on our proposed loss objectives to achieve unlearning.

---

> > ### Comment · Reviewer_xvNv · 2024-12-03
> > **Response to authors**
> >
> > Thanks for your clarifications. Most of the contents in authors' comments seem adequate to me, besides two more questions:
> > 1. The updated theoretical justification is somewhat degenerate. It is the key to involve the dynamics of data unlearning, otherwise the case is basically existing and not interesting.
> > 2. I am still not clear about how the superfactor is "auto-tuned". Are there any systematic strategies (to force a fixed portion of NegGrad gradients)?

---

> > > ### Author Response · Authors · 2024-12-04
> > >
> > > We are happy to clarify further!
> > >
> > > 1. You are correct that the standard descent lemma is well-known. However, the modified version we present under perturbations is needed to justify our use of gradient clipping when applying SISS. The gradient clipping is achieved by adjusting the superfactor $s$ in the manner described below.
> > >
> > > 2. We provide a concrete example of the superfactor tuning. Recall the SISS loss (Eq. 8 in the paper):
> > > $$
> > > E_{p_X(x)} E_{p_A(a)} E_{q_\lambda\left(m_t| x, a\right)} \Bigg[\frac{n}{n-k} \frac{q\left(m_t| x\right)}{(1-\lambda) q\left(m_t| x\right)+\lambda q\left(m_t| a\right)}\left\|\frac{m_t-\gamma_t x}{\sigma_t}-\epsilon_\theta(m_t, t)\right\|_2^2
> > > $$
> > >
> > > $$
> > > -(1+s)\frac{k}{n-k} \frac{q\left(m_t| a\right)}{(1-\lambda) q\left(m_t| x\right)+\lambda q\left(m_t| a\right)}\left\|\frac{m_t-\gamma_t a}{\sigma_t}-\epsilon_\theta(m_t, t)\right\|_2^2\Bigg]
> > > $$
> > >
> > > The first term in the loss corresponds to naive deletion, and the second term corresponds to NegGrad. Upon sampling $x$, $a$, and $m_t$, we run backpropagation to obtain gradients for both terms. Let $g_1$ be the gradient of the naive deletion term, and $g_2$ be the gradient of the NegGrad term.  Notice that adjusting $s$ changes $\|g_2\|$. Hence, we can tune $s$ so that
> > >
> > > $$
> > > \|g_2\|=p\|g_1\|
> > > $$
> > >
> > > where $p<1$ is some proportion fixed throughout unlearning in order to keep the NegGrad gradient small wrt. naive deletion gradient. In practice, we found $p=0.1$ to work well in preventing exploding gradient issues as mentioned in line 303 of the paper.
> > >
> > > We now explain the connection this procedure has to our version of the descent lemma in the PDF we linked previously (https://drive.google.com/file/d/1B1rDuibatkhVp33ducUGKeSi9m69-lSp/view?usp=sharing). In the lemma’s notation, let $f$ be the naive deletion term. $p$ will be the same as just defined (i.e., $0.1$ in practice). Finally, $v$ is set to the $g_2$ obtained after tuning $s$. Notice that the tuning of $s$ was needed to satisfy the $\|v\|\le p\|\nabla f(x)\|$ constraint in the lemma where $\nabla f(x)=g_1$ here. It is through this instantiation of the descent lemma that we conclude SISS theoretically prevents quality degradation for small enough step sizes since an improvement in $f$ (naive deletion) indicates model quality is preserved.
> > >
> > > We greatly appreciate your thoughtful consideration of our clarifications! If they have been satisfactory, we kindly ask incorporating it into your review.

---

### Official Review · Reviewer_SnfJ · 2024-11-04

**Soundness:** 3
**Presentation:** 3
**Contribution:** 3
**Rating:** 6
**Confidence:** 3

**Summary:**

The paper proposes Subtracted Importance Sampled Scores (SISS), a family of loss functions that integrate importance sampling to facilitate data unlearning while preserving model quality. The method provides theoretical guarantees for data unlearning and is experimentally evaluated on CelebA-HQ and MNIST.

**Strengths:**

1) The paper introduces an approach to data unlearning by applying importance sampling within the loss function framework, which has not been explored in the literature so far.
2) The paper is well written and structured.

**Weaknesses:**

1)  While SISS is shown to be more efficient than retraining, the paper could provide more details for the computational cost of the proposal.
2) Some important details regarding the experimental resutls are not mentioned clearly (please see the questions below).

**Questions:**

1) What is the computational cost of the proposed method? It would be helpful to have a table that summarizes the computationa cost compared to other works. For example, it is mentioned for SISS (no IS) that "Note, however, that it requires two forward passes through the denoising network ϵθ and is thus doubly more expensive in compute and memory." but it is also mentioned that the proposal is more computationally efficient than retraining.
2) Table 2 presetns results for various steps but table 1 contains results for 60 fine-tuning steps. Could you please explain the rational behind this choice? What would be the results in talbe 2 if the same number of steps were used across all the methods?
3) how does the method behave for lambda between 0 and 0.5 and between 0.5 and 1, and what does this means conceptually?

Minor comment:on page 8, there is a lot of white space that could be used or eliminated

---

> ### Author Response · Authors · 2024-12-01
>
> We appreciate the reviewer’s assessment that our paper is “well written and structured” and our approach “has not been explored in the literature so far.”
>
> **Weaknesses:** Fully addressed by questions.
>
> **Questions:**
>
> 1. With regards to retraining, it is very expensive (e.g., in Table 2 retraining requires 117500 steps) since it does not start from a pre-trained checkpoint. All other unlearning methods we test start from a pre-trained checkpoint, and the computational cost among them can be measured through the number of forward passes in the loss function. As a concrete example, this is the SISS loss (Eq. 5 in the paper):
>
> $$
> E_{p_X(x)} E_{p_A(a)} E_{q_\lambda\left(m_t| x, a\right)} \Bigg[\frac{n}{n-k} \frac{q\left(m_t| x\right)}{(1-\lambda) q\left(m_t| x\right)+\lambda q\left(m_t| a\right)}\left\|\frac{m_t-\gamma_t x}{\sigma_t}-\epsilon_\theta(m_t, t)\right\|_2^2
> $$
>
> $$
> -\frac{k}{n-k} \frac{q\left(m_t| a\right)}{(1-\lambda) q\left(m_t| x\right)+\lambda q\left(m_t| a\right)}\left\|\frac{m_t-\gamma_t a}{\sigma_t}-\epsilon_\theta(m_t, t)\right\|_2^2\Bigg]
> $$
>
> Notice there is only one forward pass through the denoising network due to the term $\epsilon_\theta(m_t, t)$.
>
> However, in the SISS (No IS) loss function (Eq. 4 in the paper),
>
> $$
> \frac{n}{n-k} E_{p_X(x)} E_{q(x_t|x)}\left\|\frac{x_t-\gamma_t x}{\sigma_t}-\epsilon_\theta(x_t, t)\right\|_2^2
> $$
>
> $$
> -\frac{k}{n-k} E_{p_A(a)} E_{q(a_t|a)}\left\|\frac{a_t-\gamma_t a}{\sigma_t}-\epsilon_\theta(a_t, t)\right\|_2^2.
> $$
>
> we observe there are two forward passes due to the terms $\epsilon_\theta(x_t, t)$ and $\epsilon_\theta(a_t, t)$ making it twice as computationally expensive.
>
>
> 2. We agree that it is inconsistent to have shared timesteps for CelebA-HQ (Table 1) but not for MNIST T-Shirt (Table 2). Here are the updated numbers of the red methods in Table 2 after 300 fine-tuning steps:
>
> ### MNIST T-Shirt Results (Table 2 in Paper)
>
> | Method              | Steps | IS   | NLL    | p   |
> |---------------------|-------|-------|--------|-----|
> | EraseDiff           | 300   | 2.2   | 15.16  | N/A |
> | SISS (λ = 1.0)      | 300   | 1.1   | 13.34  | N/A |
> | NegGrad             | 300   | 1.0   | 20.23  | N/A |
>
> The main difference is that the quality metric (Inception Score) for those red methods is significantly worse. They have higher NLL but that is not very meaningful if the model quality is already very poor.
>
> 3. Here are additional results for $\lambda=0.25, 0.75$:
> ### CelebA-HQ (Table 1 in Paper)
>
> | Method              | FID  | NLL   | SSCD |
> |---------------------|-------|-------|------|
> | SISS ($λ = 0.25$)     | 24.6  | 1.433 | 0.39 |
> | SISS ($λ = 0.5$)      | 25.1  | 1.442 | 0.36 |
> | SISS ($λ = 0.75$)     | 27.7  | 1.454 | 0.36 |
>
> ### MNIST T-Shirt (Table 2 in Paper)
>
> | Method              | Steps | IS   | NLL   | p   |
> |---------------------|-------|-------|-------|-----|
> | SISS ($λ = 0.25$)     | 300   | 9.3   | 7.98  | 0%  |
> | SISS ($λ = 0.5$)      | 300   | 9.2   | 8.09  | 0%  |
> | SISS ($λ = 0.75$)     | 300   | 9.1   | 8.18  | 0%  |
>
> As long as $\lambda$ is not too far from $0.5$, the results will be similar. Generally, the objective of IS is to reduce estimator variance. $\lambda$, in particular, controls the tradeoff in variance between naive deletion and NegGrad. If $\lambda$ is close to $1$, then we will have a good estimate for NegGrad but a poor one for naive deletion and vice versa for $\lambda\approx 0$.
>
> The extra whitespace on page 8 will be fixed in the camera-ready version of the paper.

---

### Public Comment · ~Kyeonghyun_Lee1 · 2024-11-29
**Questions on implementation of loss functions**

I have some questions about the provided code.

In your code, there are various losses defined in ddpm_deletion_loss.py and we have tested them. However, I can't understand how they match to the losses provided in Table 1. Can you explain how they are matched?

---

> ### Public Comment · ~Kenan_Hasanaliyev1 · 2025-03-02
> **Updated codebase and improved clarity**
>
> Thanks for pointing out the lack of clarity! We have updated the codebase and uploaded the latest version to https://github.com/claserken/SISS. The updated ddpm_deletion_loss.py specifies the matching to unlearning methods in our paper.

---

### Meta-Review · Area_Chair_cskG · 2024-12-23

**Metareview:**

The paper proposes Subtracted Importance Sampled Scores (SISS), a family of loss functions that integrate importance sampling to facilitate data unlearning while preserving model quality.  SISS is constructed as a weighted combination between simpler objectives responsible for preserving model quality and unlearning the targeted data points. The paper shows strong empirical results on CelebA-HQ, MNIST T-Shirt, and Stable Diffusion. The reviewers generally find the method proposed simple, the empirical results promising, and the paper well-written. Meanwhile, some reviewers believe this work will benefit from more theoretical justification/intuition for the proposed method.

**Additional Comments On Reviewer Discussion:**

3 out of 4 reviewers leaned toward acceptance. Reviewer JqNx gave a score of 3 and had some questions about the idea and details of the proposed method. The AC thinks these questions were addressed satisfactorily and does not see any significant concerns remaining.

---

### Decision · Program_Chairs · 2025-01-22

Accept (Poster)